# Spatial disparities and multilevel determinants of childhood diarrhea in Mozambique: Evidence from the 2022–2023 Demographic and Health Survey (DHS)

**Thomas Kidanemariam Yewodiaw**[1,2*], **Mihret Getnet**[2,3], **Mequanent Dessie Bitewa**[4], **Hiwot Tezera Endale**[5]

**1** Medical Officer at International Medical Corps, Amhara Region Emergency Operation Center, Gondar Field Office, Gondar, Ethiopia, **2** Department of Epidemiology and Biostatistics, Institute of Public Health, College of Medicine and Health Sciences, University of Gondar, Gondar, Ethiopia, **3** Department of Human Physiology, School of Medicine, College of Medicine and Health Sciences, University of Gondar, Gondar, Ethiopia, **4** Department of Public Health, College of Health Sciences, Debre Markos University, Debre Markos, Ethiopia, **5** Department of Medical Biochemistry, School of Medicine, College of Medicine and Health Sciences, University of Gondar, Gondar, Ethiopia

\* thomaskmariam28@gmail.com

## Abstract

### Background

Childhood diarrhea remains a major public health problem, particularly in sub-Saharan Africa, contributing substantially to morbidity and mortality and hindering progress toward Sustainable Development Goal 3 (SDG 3), which aims to end preventable under-five deaths. Marked regional variations in diarrhea burden highlight the need for updated analyses to guide public health planning and resource allocation. This study examines the spatial distribution and key determinants of diarrhea among under-five children to inform targeted interventions.

### Methods

Data from 9,799 under-five children in the 2022–2023 Mozambique Demographic and Health Survey were analyzed to estimate diarrhea prevalence and identify associated determinants. Weighted analysis, spatial scan statistics (SaTScan), hotspot mapping, and multilevel logistic regression were used to assess individual- and community-level factors. Significant predictors were identified using adjusted odds ratios (AORs) with 95% confidence intervals (CIs) and $p \leq 0.05$. Model variation was assessed using the intra-class correlation coefficient (ICC), median odds ratio (MOR), and proportional change in variance (PCV), and spatial patterns were mapped using ArcGIS.

**Data availability statement:** The data used in this study are publicly available from the Demographic and Health Surveys (DHS) Program. The dataset for Mozambique 2022/2023 can be requested at https://dhspro-gram.com/data/available-datasets.cfm.

**Funding:** The author(s) received no specific funding for this work.

**Competing interests:** No competing interest.

**Abbreviations:** U5C, Under-Five Children; SaTScan, Spatial and Space-Time Scan Statistics; GPS, Global Positioning System; WHO, World Health Organization; SSA, Sub-Saharan Africa; UNICEF, United Nations Children's Fund; ICC, intraclass correlation coefficient; MOR, median odds ratio; PCV, proportional change in variance; MDHS, Mozambique Demographic Health Surveys; KR, Kids Recode (DHS child-level dataset); EAs, enumeration areas; AOR, Adjusted Odds Ratio; CI, Confidence Interval; p-value, Probability value (significance level).

## Results

The weighted prevalence of diarrhea among under-five children in Mozambique was 8.8% (95% CI: 7.8–9.6%), highest in Niassa (14.5%), Cabo Delgado (14.0%), and Maputo City (13.4%), and lowest in Maputo Province (5.1%), Zambézia (5.5%), and Manica (5.8%). Multilevel analysis showed that children aged 12–23 months had higher odds of diarrhea (AOR = 1.36, 95% CI: 1.11–1.66). In contrast, children aged 24–59 months (AOR = 0.49, 95% CI: 0.41–0.59), maternal education (AOR = 0.77, 95% CI: 0.63–0.96), and rural residence (AOR = 0.69, 95% CI: 0.52–0.91) were associated with lower odds. Children residing in Nampula, Zambézia, Manica, Sofala, and Maputo Province had significantly lower odds of diarrhea compared to those in Niassa. Health-seeking behavior was strongly associated with reported diarrhea (AOR = 4.85, 95% CI: 4.10–5.74), possibly reflecting reporting bias.

## Conclusions

Childhood diarrhea in Mozambique exhibits marked regional variation, with the highest burden in Niassa, Cabo Delgado, and Maputo City, and the lowest in Maputo Province, Zambézia, and Manica. Key determinants include child age, maternal education, and geographic region, while rural residence appears protective. These findings highlight the need for targeted, age- and region-specific interventions and strengthened maternal and community support to reduce childhood diarrhea and accelerate progress toward SDG 3.

## Introduction

Diarrheal disease is the third leading cause of death in children 1–59 months of age, particularly in low-income and rural settings. However, it is preventable and treatable [1]. According to recent WHO and UNICEF estimates (2024–2025), diarrheal diseases account for approximately 444,000 deaths among children under five annually, compared to about 390,000 deaths reported in 2021, highlighting a persistent global health burden [1–5]. Nearly 1.7 billion cases of childhood diarrhea occur every year worldwide [1].

The prevalence of diarrhea among children under five years old in East Africa and Sub-Saharan countries was 14.28% and 15.3% respectively [6,7]. The 2011 Mozambique demographic and health survey revealed an 11% prevalence of childhood diarrhea, with higher rates in Tete and Zambia, but lower rates in Gaza and Cabo Delgado [8]. UNICEF (2021) highlights diarrhea as a major cause of under-five mortality in Mozambique, exacerbated by inadequate access to clean water, poor sanitation, and limited healthcare services [9]. In Mozambique, diarrhea continues to be a moderate but persistent public health concern, especially among children under five [10]. Previous surveys have revealed geographical inequalities, with higher frequency in northern regions, frequently connected to insufficient water and sanitation, low maternal education, and limited healthcare access [11,12]. The inequalities

underscore the need to analyze both individual-level factors like child age and maternal employment and community-level determinants like region and rural/urban setting [13].

Childhood diarrhea is influenced by factors at multiple levels, including individual (age, breastfeeding, vaccination, nutrition), household (size, caregiving, maternal age and education), and environmental (water source, sanitation, socio-economic status) [14–18].

DHS data-driven spatial epidemiological analyses help identify high-risk geographic clusters for childhood diarrhea, enabling policymakers to allocate resources and implement intervention strategies in high-burden areas [19]. Mozambique, a southeastern African country, is significantly burdened by childhood diarrhea, with high prevalence in children under five, varying across provinces. In alignment with SDG Goal 3, Mozambique has committed to reducing under-five mortality to 25 per 1,000 live births [20,21]. The 2022/2023 Mozambique Demographic and Health Survey (DHS) offers recent, nationally representative data to reassess diarrhea prevalence and patterns across the region [10,22]. The analysis of recent public health initiatives and changes in infrastructure and health services is crucial to identify changes in disease distribution and risk factors [20]. In Mozambique, under-five diarrhea has historically been a major contributor to child mortality, with significant variations across regions. Efforts to reduce diarrhea-related deaths have included the introduction of rotavirus vaccination, improvements in water, sanitation, and hygiene (WASH) infrastructure, expansion of trained healthcare personnel, and implementation of national child health guidelines. These interventions have contributed to modest declines in diarrhea incidence and mortality; however, substantial geographic and socio-demographic disparities persist. Childhood diarrhea is a major contributor to under-five morbidity and mortality worldwide, particularly in sub-Saharan Africa, where progress toward SDG 3 remains uneven [23]. In Mozambique, national and subnational estimates indicate significant regional disparities in diarrhea prevalence, highlighting the need for targeted, evidence-based interventions [7,24,25]. Understanding spatial patterns and multilevel determinants of childhood diarrhea is essential for designing targeted interventions and advancing Mozambique's progress toward Sustainable Development Goal 3, which aims to end preventable under-five deaths. This study therefore examines the spatial distribution and multilevel determinants of diarrhea among children under five using the 2022–2023 DHS and spatial modeling tools such as SaTScan and ArcGIS to inform evidence-based public health strategies.

## Methods and materials

### Study design and study setting

This study is a cross-sectional secondary data analysis of the 2022/2023 Mozambique Demographic and Health Survey. Mozambique, located in south-east Africa at 18° 15' South and 35° 00' East, has an estimated 34 million people, with the majority residing in rural areas [26,27]. The country borders the Indian Ocean, Tanzania, Malawi, Zambia, Zimbabwe, Eswatini, and South Africa. Mozambique is divided into 10 provinces and one city province (Maputo City) [28].

### Data source and sampling procedure

The analysis utilizes Mozambique 2022/2023 Demographic and Health Survey data (DHS) sourced from the MEASURE DHS public repository. The study utilized the latest data set from the DHS, a worldwide survey conducted every five years in low- and middle-income nations. The data was gathered from a national representative sample of approximately 9,799 households in all 11 regions of Mozambique. The 2022/2023 Mozambique Demographic and Health Survey employed stratified two-stage cluster sampling to yield representative results at the national level for both urban and rural areas. The 619 targeted clusters were selected using a probability-proportional-to-size strategy for both urban and rural areas, followed by systematic random sampling to ensure equal probability. In the second stage, household listings were updated, and maps in all selected clusters were created to create a list of households for each cluster. The household sample was drawn from the list. From the list, a predetermined number of 30 households per cluster were chosen at random

to participate in interviews. A total of 9,799 children aged 0–59 months from 619 clusters were included in the analysis. These children were drawn from households in both rural and urban areas. Information on child health was obtained from interviews conducted with mothers or primary caregivers. In cases where mothers had more than one eligible child, data were collected for the most recent child.

## Study population

A total weighted sample of 9,799 children aged 0–59 months from 619 clusters was included in the analysis. These children were drawn from households in both rural and urban areas across Mozambique. Information on children was obtained from interviews conducted with their mothers or primary caregivers. In cases where mothers had more than one eligible child, data were collected for the most recent child born within the five years preceding the survey.

## Inclusion and exclusion criteria

The study included children aged 0–59 months who were part of the selected enumeration areas (EAs) in the 2022/2023 Mozambique Demographic and Health Survey. Children with missing information on diarrheal status during the two weeks preceding the survey were excluded from the analysis.

## Study variables

**Outcome of variable.** The outcome variable for this study was the occurrence of diarrhea among children aged 0–59 months. It was defined based on maternal report of whether the child had experienced diarrhea during the two weeks preceding the survey. The variable was coded as "Yes = 1" if the child had diarrhea and "No = 0" otherwise.

**Independent variables.** Those factors were reviewed from different literature, including child age, breastfeeding status, vaccination status, nutritional status, household size, caregiving behavior, maternal age, education level, residence, region, water source, sanitation, and socioeconomic status [14–18].

## Operational definitions

**Diarrhea:** A child is considered to have diarrhea if the mother or caregiver reports that the child passed three or more loose or watery stools in 24 hours within the two weeks preceding the survey [22].

**Media Access:** A respondent is considered to have media access if they report accessing any one of these media sources at least occasionally (newspapers or magazines, radio, or television) [22].

**Types of toilets:** Toilets such as flush to elsewhere, pit latrines without slabs, bucket, hanging, or other types were classified as **unimproved**. In contrast, flush toilets connected to sewers, septic tanks, pit latrines, ventilated or slab pit latrines, and composting toilets were considered **improved sanitation facilities [22,29,30]**.

**Source of Water:** Drinking water sources are classified as **improved** if they include piped water, public taps, tube wells, protected wells or springs, rainwater, and bottled water. Sources like unprotected wells or springs, tanker trucks, surface water, and bottled water used alone are considered **unimproved [22,29]**.

**Wealth Index** is a combined measure of a household's overall living conditions, based on ownership of assets (TV, bicycle), housing quality (floor material, water source, sanitation facilities), and access to services. Using principal components analysis, DHS assigns weights to these factors and classifies households into five groups, ranging from poorest to richest [22,31].

## Data management and analysis

The study utilized data from various sources, including median, table, and percent, to analyze key characteristics and estimate response rates and frequencies due to non-proportional sample allocation. The weighting procedure can be

explained in more detail on the DHS portal at https://dhsprogram.com/data/Guide-to-DHSStatistics/Analyzing_DHS_Data.htm. Figures were created using ArcGIS version 10.7 software, using Mozambique regional shape file for hotspot area, kriging interpolation, and SatScan windows, available at https://data.humdata.org/dataset/cod-ab-moz

## Spatial analysis

The spatial analysis of under-five diarrhea in Mozambique was conducted using ArcGIS 10.7 and Sat Scan version 9.6, using Global Moran's I statistics measure to evaluate its distribution (dispersed, clustered, or random). Moran's I value is a spatial statistic used to measure spatial autocorrelation by generating a single value from −1–1. Moran's, I value indicates a clustered pattern of under-five diarrhea; a negative value indicates a dispersed pattern, and a value close to zero indicates a random distribution [32,33]. All spatial analyses were conducted at the cluster level using DHS survey clusters as the spatial unit. The GPS coordinates provided for each cluster (randomly displaced to protect participant confidentiality) were used to: Identify high-risk clusters of diarrhea cases using Gi* hotspot analysis. Assess residual spatial autocorrelation after multilevel modeling with Moran's I. We did not perform spatial analysis at the individual child level because individual GPS locations are unavailable, and the cluster-level approach ensures both privacy protection and statistical robustness. Analyses accounted for the DHS complex survey design, including sampling weights, clusters, and strata. Multilevel logistic regression included random intercepts for clusters and provinces to capture hierarchical clustering, while spatial analyses at the cluster level identified high-risk hotspots and assessed spatial autocorrelation. A spatially lagged diarrhea prevalence variable was included in the multilevel model to account for residual spatial dependence, ensuring both the hierarchical and spatial structure of the data were addressed. All maps were generated by the authors using DHS cluster GPS data and open-access administrative boundary shapefiles obtained from the Humanitarian Data Exchange (HDX), without the use of proprietary base maps or satellite imagery, and are compliant with CC BY 4.0 licensing."

**Getis-Ord Gi* statistics hotspot analysis [34]** was used to identify the hotspots and cold spots in the cluster, indicating a higher or lower proportion of under-five diarrhea illness. Hotspot and cold spot clusters are indicated by high and low proportions of diarrhea among children under five children respectively. The spatial interpolation technique was employed to predict diarrheal illnesses among children under five in unsampled areas based on sampled EA measurements. This study utilized the ordinary Kriging spatial interpolation method, which has the smallest root mean square error value and residuals, to predict diarrheal illnesses in unobserved areas [35]. The study utilized spatial scan statistical analysis (SaTScan) with the Bernoulli distribution to identify significant spatial clusters of childhood diarrheal illnesses using Kulldorf's SaTScan V.9.6 software [36].

## Multilevel analysis

The multilevel multivariable logistic regression model was used to analyze the association between individual and community-level factors, with fixed effect estimates and a 95% confidence interval. Multilevel analysis is appropriate for nested data, such as DHS. Individual-level factors include child age, sex, mother's education, mother's employment, mother age, household wealth, Household size, and nutritional status are associated risk of diarrhea among children. However, Mothers' individual characteristics, like as income and education, may be influenced by community-level factors such as place, media exposure, poverty, and region. Individual-level characteristics of children in DHS data tend to be more connected within clusters than between other clusters. The similarity of individual characteristic scores within a cluster violates conventional regression's independence assumptions. Multilevel analysis can overcome the lack of independence of observations in nested data analysis. A two-level binary logistic regression (individual and community level) was used to determine risk variables for diarrhea.

A total of four models were fitted. The null model, commonly known as the random intercept model, was used to assess cluster variability in diarrhea. Model fit was determined using fitness requirements such as the Likelihood Ratio test (LLR),

deviance, Akaike information criterion (AIC), and Deviance Information Criterion (DIC). The model with the lowest fitness parameters was chosen as the optimal fit. The study evaluated cluster variability using the intra-class coefficient (ICC) [2] and median odds ratio and Proportional Change in Variance (PCV). The ICC measures the percentage variation in community-level variables, while PCV measures the proportional change in community-level variance between null and succeeding models [37]. The Median Odds Ratio (MOR) is a statistical measure that measures the area-level variance of the odds ratio scale. It is calculated by comparing the median value of the odds ratio between high and low risk areas. In the absence of area-level variation, the MOR is equal to 1. Adjusted odds ratio with 95% confidence interval (CI) and p-value < 0.05 were used to declare statistical significance.

**Ethical considerations**This study was a secondary analysis of the 2022–2023 Mozambique Demographic and Health Survey (DHS). The original survey protocol, including the data collection tools and procedures, was reviewed and approved by the Institutional Review Board (IRB) of ICF and the Mozambique National Committee for Bioethics in Health (CNBS). During the primary survey, written informed consent was obtained from all adult participants; for children included in the study, consent was obtained from their parents or legal guardians.

The authors obtained authorization to use the dataset from the DHS Program on February 2, 2024. The data provided to the authors were fully anonymized and de-identified by the DHS Program to ensure participant confidentiality. Consequently, this study was exempt from further institutional ethical review. The datasets are available to registered users through the DHS Program website (https://dhsprogram.com).

## Results

### Characteristics of the study population

The majority of under-five children were aged 24–59 months, accounting for 60.0% (5,881) of the sample. Females slightly outnumbered males at 51.4% (5,032) compared to 48.6% (4,767). Most mothers had a primary education, 49.1% (4,810), while 30.1% (2,950) had no formal education. The dominant maternal age group was 20–34 years, 69.5% (6,812). A large proportion of mothers were not working, 73.2% (7,172). The sample was mostly rural, 71.3% (6,991). In terms of wealth, 48.1% (4,709) of children were from low-income households. Nearly half lived in small households, 48.8% (4,783). 94.0% (n = 9,214) of children were classified as having normal nutritional status, while underweight cases were 6.0% (585). Only 25.3% (2,483) of children were fully vaccinated. Most children lived in communities with low underweight prevalence (98.7%, n = 9,670), while only a small proportion resided in high-prevalence communities. In terms of community education level, the majority resided in medium (37.5%; n = 3,673) or high (35.2%; n = 3,452) education-level communities (Table 1).

### National prevalence of diarrhea among under-five children

The overall prevalence of diarrhea among children in the 2022/2023 Mozambique DHS KR dataset was 8.8% (95% CI: 7.8% to 9.6%). Diarrhea was more common among children aged 12–23 months, with a prevalence of 15.1% (95% CI: 13% to17.5%), and was more frequently reported among urban (11.7%). The burden was higher among children of mothers with secondary education and above (10.9%) (Table 2).

### Regional variation in diarrhea prevalence

The 2022/2023 DHS data reveal that the prevalence of diarrhea among children in Mozambique shows noticeable regional variation. The two provinces with the **highest diarrhea prevalence** are **Niassa** (14.6%) and **Cabo Delgado** (14.1%). In contrast, the **lowest prevalence** is observed in **Maputo Province** (5.2%) and **Zambézia** (5.5%) (Table 2).

**Table 1. Characteristics of the study population of children under five in Mozambique, DHS 2022/2023.**

| Variable | Category | Weighted Frequency | Percentage (%) |
|---|---|---|---|
| **Age group (months)** | 0–11 | 2,018 | 20.6 |
| Age group (months) | 12–23 | 1,900 | 19.4 |
| Age group (months) | 24–59 | 5,881 | 60.0 |
| **Sex** | Male | 4,767 | 48.6 |
| Sex | Female | 5,032 | 51.4 |
| **Maternal education** | No education | 2,950 | 30.1 |
| Maternal education | Primary | 4,810 | 49.1 |
| Maternal education | Secondary+ | 2,039 | 20.8 |
| **Mother age group** | <20 | 1,036 | 10.6 |
| Mother age group | 20–34 | 6,812 | 69.5 |
| Mother age group | ≥35 | 1,951 | 19.9 |
| **Maternal work status** | Working | 2,627 | 26.8 |
| Maternal work status | Not working | 7,172 | 73.2 |
| **Residence** | Urban | 2,808 | 28.7 |
| Residence | Rural | 6,991 | 71.3 |
| **Wealth index** | Low | 4,709 | 48.1 |
| Wealth index | Medium | 1,943 | 19.8 |
| Wealth index | Higher | 3,148 | 32.1 |
| **Household size** | Low | 4,783 | 48.8 |
| Household size | Medium | 2,756 | 28.1 |
| Household size | High | 2,260 | 23.1 |
| **Nutritional status** | Normal | 9,214 | 94.0 |
| Nutritional status | Underweight | 585 | 6.0 |
| **Community underweight** | Low | 9,670 | 98.7 |
| Community underweight | High | 129 | 1.3 |
| **Community poverty** | Low | 2,725 | 27.8 |
| Community poverty | Medium | 7,074 | 72.2 |
| **Community education level** | Low | 2,674 | 27.3 |
| Community education level | Medium | 3,673 | 37.5 |
| Community education level | High | 3,452 | 35.2 |

## Spatial analysis

**Spatial autocorrelation.** The geographical autocorrelation analysis indicated regional variation in diarrhea among children in Mozambique. The Global Moran's I was 0.05, with a Z-score of 1.74 and a p-value of 0.08. While these values suggest a weak positive spatial autocorrelation, the result is not statistically significant at the 5% level. Therefore, although there may be a slight tendency toward geographic clustering, we cannot confidently conclude that the spatial distribution of diarrhea is non-random. Further localized spatial analysis is recommended to explore potential clustering at sub-regional levels (Fig 1).

**High/Low hotspot analysis.** The hotspot analysis using the Getis-Ord General G statistic revealed a Z-score of 1.95, a p-value of 0.05, and an observed General G value of 0.000. These results suggest that there is moderate but statistically significant clustering of high values across the study area. Since the p-value is exactly 0.05, this indicates that the spatial clustering is just statistically significant at the 5% level, and the positive Z-score points to the presence of hotspots

**Table 2. Weighted Prevalence of Diarrhea Among Under-Five Children by Selected Characteristics (Mozambique DHS 2022/2023.**

| Variable | Category | Weighted N | Diarrhea Prevalence % (95% CI) |
|---|---|---|---|
| **Child age (months)** | 0–11 | 202 | 10.4 (8.7–12.3) |
| Child age (months) | 12–23 | 273 | 15.1 (13.0–17.5) |
| Child age (months) | 24–59 | 342 | 6.1 (5.3–7.0) |
| **Sex of child** | Male | 385 | 8.5 (7.4–9.7) |
| Sex of child | Female | 432 | 8.9 (7.9–10.1) |
| **Place of residence** | Urban | 315 | 11.7 (10.2–13.4) |
| Place of residence | Rural | 502 | 7.5 (6.5–8.6) |
| **Maternal working status** | Working | 278 | 10.9 (9.4–12.6) |
| Maternal working status | Not working | 539 | 7.9 (6.9–8.9) |
| **Community poverty** | Low | 289 | 11.0 (9.6–12.4) |
| Community poverty | Medium | 528 | 7.8 (6.8–9.0) |
| **Province (Region)** | Niassa | 116 | 14.6 (11.7–18.0) |
| Province (Region) | Cabo Delgado | 86 | 14.1 (11.3–17.5) |
| Province (Region) | Cidade de Maputo | 29 | 13.4 (10.1–17.7) |
| Province (Region) | Tete | 121 | 12.3 (8.7–16.9) |
| Province (Region) | Gaza | 43 | 12.0 (9.7–14.9) |
| Province (Region) | Inhambane | 31 | 11.0 (8.1–14.8) |
| Province (Region) | Sofala | 66 | 10.4 (8.3–12.8) |
| Province (Region) | Nampula | 161 | 6.5 (4.9–8.6) |
| Province (Region) | Manica | 42 | 5.9 (4.3–7.9) |
| Province (Region) | Zambézia | 96 | 5.5 (3.9–7.7) |
| Province (Region) | Maputo Province | 26 | 5.2 (3.2–8.3) |

areas where high values of the studied variable are more concentrated than would be expected by chance. Although the observed General G value is close to zero, the Z-score and p-value together imply that there is a tendency for high values to be spatially clustered. Further local spatial analysis, such as the Getis-Ord Gi* statistic, is recommended to pinpoint specific hotspot locations within the country (Fig 2).

**Hotspot analysis.** Hotspot analysis was performed to identify high-risk areas of under-five diarrhea in Mozambique. The red color (hotspot) indicates significant risky areas and is found in Northern Region (Cabo Delgado, Niassa, Nampula, Zambezia) and Central Region (Tete, Manica, Sofala), whereas the blue color indicates less risky areas (cold spot) of under-five diarrhea and is observed in Southern Region (Gaza, Inhambane, Maputo Province, Maputo City) and the Coastal Region (Fig 3).

Administrative boundary shapefiles were obtained from the Humanitarian Data Exchange (HDX) (Mozambique Subnational Administrative Boundaries). All maps were produced by the authors using Demographic and Health Survey (DHS) cluster GPS data in combination with open-access spatial datasets. No proprietary base maps were used, and all spatial data sources comply with the Creative Commons Attribution (CC BY 4.0) licensing requirements.

**Spatial interpolation (Ordinary Kriging).** The spatial interpolation results from the 2022/2023 Mozambique Demographic and Health Survey (DHS) revealed that the highest predicted prevalence of diarrhea illness is located in the northern regions (Niassa and Cabo Delgado). Maputo City, found in the south, also appears as an area with high predicted prevalence. Moderate prevalence levels are observed in the central regions (Tete, Gaza, Inhambane, and Sofala). The lowest predicted prevalence of diarrhea among children under five children are detected in southern regions, especially in Maputo Province, Nampula, Manica, and Zambezia (Fig 4).

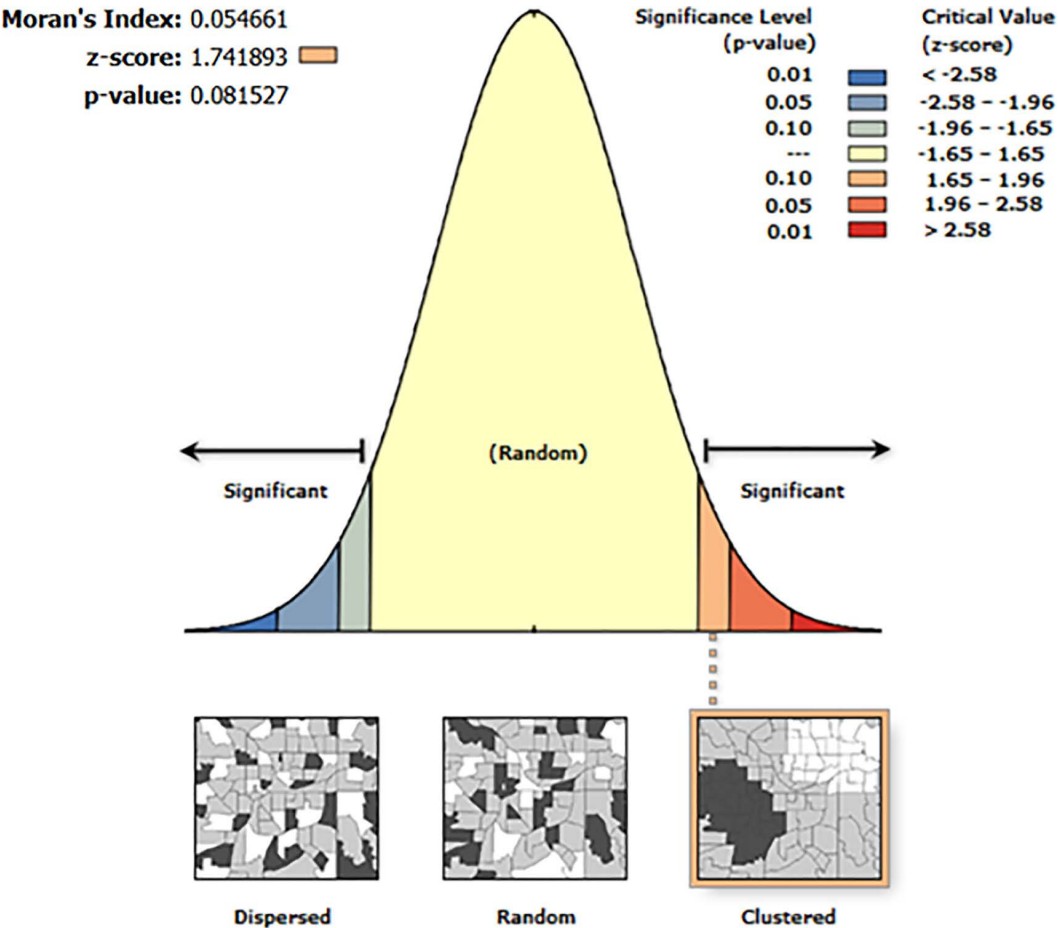

**Fig 1. Global Spatial Autocorrelation (Moran's I) of Diarrhea Prevalence Among Children Under Five in Mozambique, DHS 2022/2023.**

Administrative boundary shapefiles were obtained from the Humanitarian Data Exchange (HDX) (Mozambique Subnational Administrative Boundaries). Maps were produced by the authors using Demographic and Health Survey (DHS) cluster GPS data in combination with open-access spatial datasets. No proprietary base maps were used, and all data sources comply with Creative Commons Attribution (CC BY 4.0) licensing requirements.

### SaTScan cluster detection (Spatial Scan Statistics)

According to the SaTscan analysis, under-five diarrhea cases in Mozambique are not randomly distributed, but are concentrated in specific high-risk areas across 93 geographic areas (within two clusters), primarily in the north (Niassa and Cabo Delgado) and parts of the central and southern regions (Tete and Gaza). The first cluster includes 10 locations and is centered around coordinates (13.42°S, 39.22°E) with a radius of 41.38 km. This cluster has a relative risk of 2.79, indicating that children in this area are nearly three times more likely to experience diarrhea compared to those outside the cluster. The log likelihood ratio is 24.65, and the p-value < 0.01, confirming a highly significant clustering. The second cluster includes 83 locations and is centered around coordinates (14.59°S, 34.49°E) with a radius of 213.63 km. This cluster has a relative risk of 1.61. The log likelihood ratio is 18.47, with a p-value <0.01, also indicating strong statistical significance. From three up to seven spatial clusters, like Maputo Province, Zambezia, Manica, and Nampula, were located

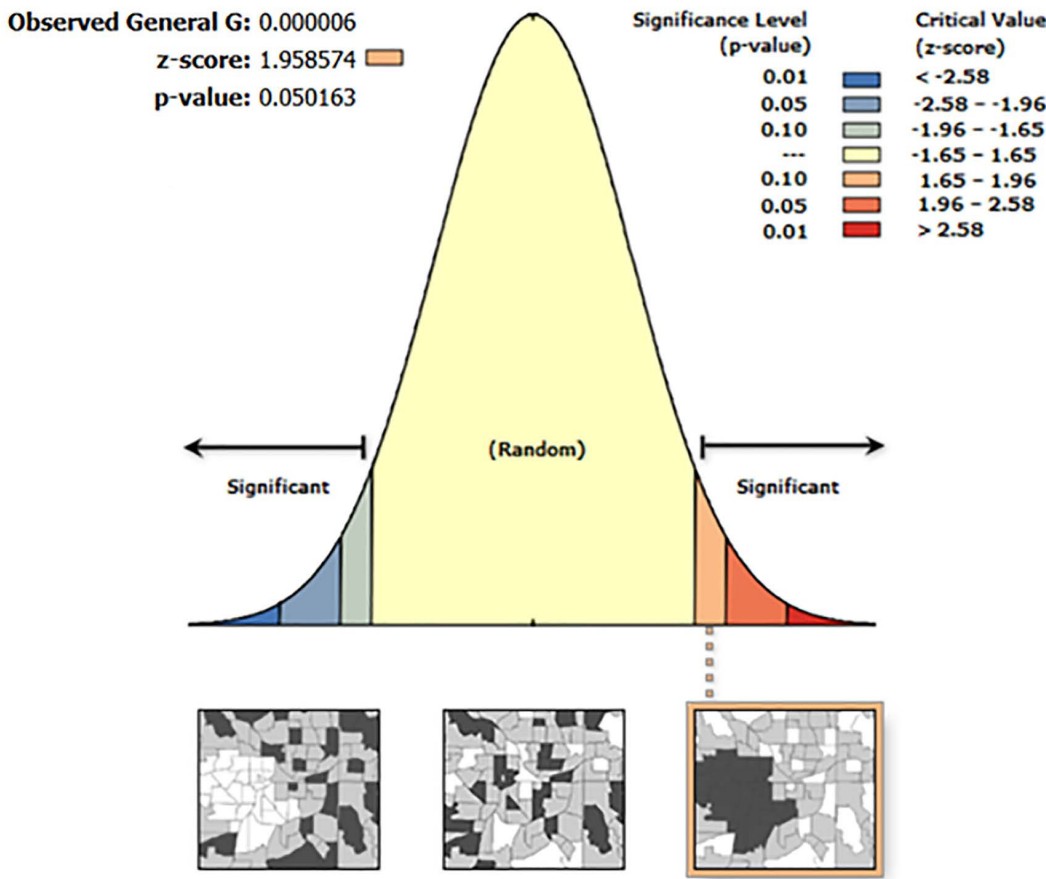

**Fig 2. Global Spatial Autocorrelation (Moran's I) of Diarrhea Prevalence Among Children Under Five in Mozambique, DHS 2022/2023.**

outside statistically significant clusters. These areas reflect a lower predicted prevalence of diarrhea among children under five (Fig 5, Table 3).

"Administrative boundary shapefiles were obtained from the Humanitarian Data Exchange (HDX): Mozambique Subnational Administrative Boundaries. Maps were created by the authors using DHS cluster GPS data and open-access spatial data, without proprietary base maps, and are compliant with CC BY 4.0 licensing."

### Individual- and community-level factors associated with diarrhea

In the fully adjusted multilevel logistic regression model (Model 3), several individual- and community-level variables were significantly associated with the outcome. Child age was significantly associated with the outcome. Compared with children aged 0–11 months, those aged 12–23 months had higher odds of the outcome (AOR = 1.36, 95% CI: 1.11–1.66), whereas children aged 24–59 months had lower odds (AOR = 0.49, 95% CI: 0.41–0.59). Children whose mothers had primary education had reduced odds of the outcome compared with those whose mothers had secondary or higher education (AOR = 0.77, 95% CI: 0.63–0.96). Households with improved health-seeking behavior had significantly higher odds (AOR = 4.85, 95% CI: 4.10–5.74) compared with those with poor health-seeking behavior.

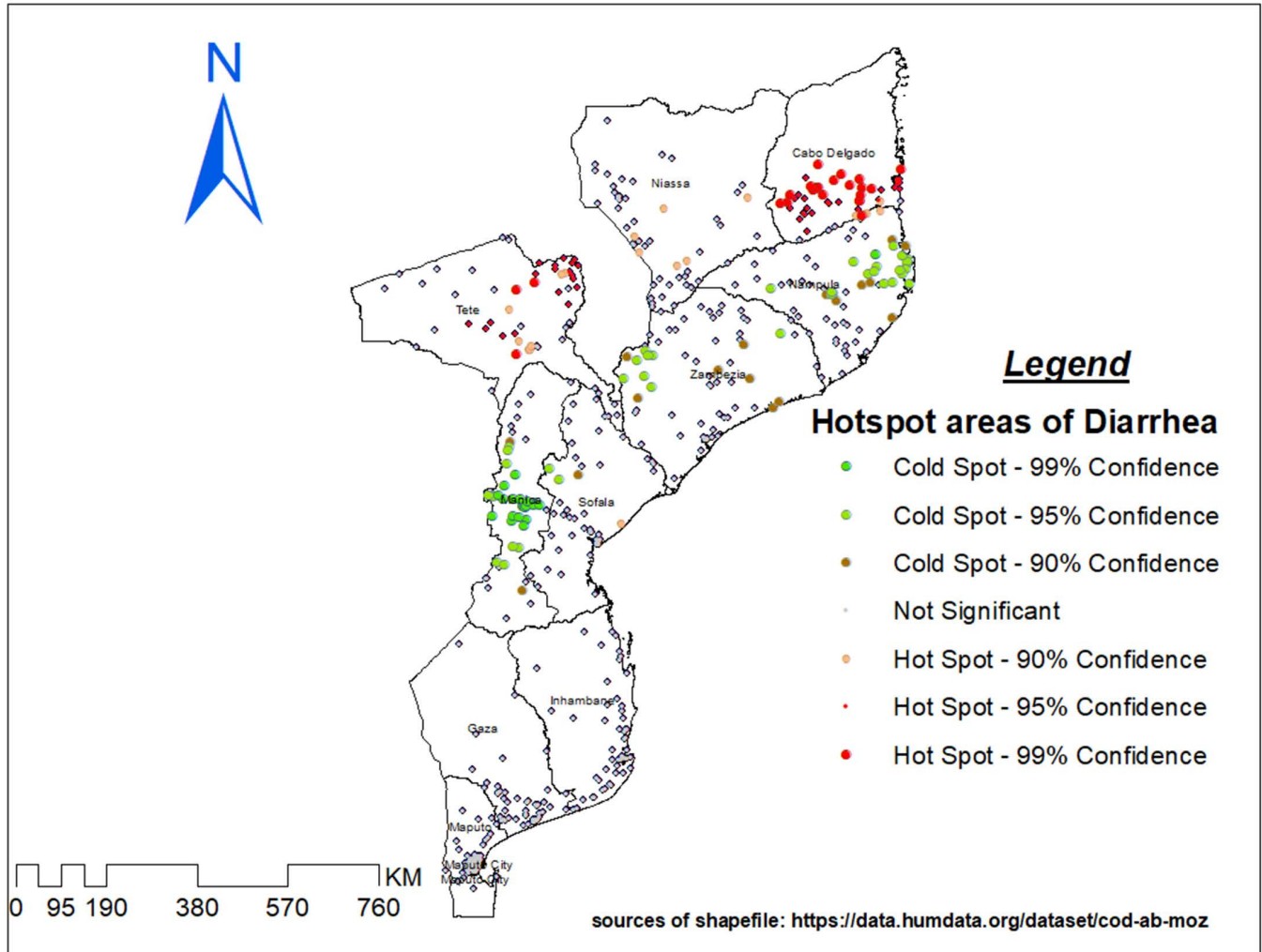

**Fig 3. High and Low Hotspot Analysis of Diarrhea Prevalence Among Under-Five Children.**

Children residing in rural areas had lower odds of the outcome compared with those living in urban areas (AOR = 0.69, 95% CI: 0.52–0.91).

Compared with Niassa Province, children residing in Nampula (AOR = 0.42, 95% CI: 0.29–0.61), Zambézia (AOR = 0.35, 95% CI: 0.23–0.55), Manica (AOR = 0.30, 95% CI: 0.20–0.45), Sofala (AOR = 0.51, 95% CI: 0.34–0.75), Inhambane (AOR = 0.49, 95% CI: 0.32–0.76), Gaza (AOR = 0.54, 95% CI: 0.35–0.82), Maputo Province (AOR = 0.26, 95% CI: 0.15–0.44), and Maputo City (AOR = 0.50, 95% CI: 0.32–0.80) had significantly lower odds of the outcome.

Regarding model fitness and random effects, the inclusion of both individual- and community-level variables reduced the cluster-level variance from 0.59 in the null model to 0.30 in the final model. The intra-class correlation coefficient (ICC) decreased from 15.2% to 8.5%, indicating a reduction in between-cluster variability. Similarly, the median odds ratio (MOR) declined from 2.08 to 1.69, suggesting decreased heterogeneity across clusters. The proportional change in variance (PCV) in the final model was 49.2%, indicating that nearly half of the variance was explained by the included variables. Model comparison showed that Model 3 had the lowest deviance (2596.5), AIC (5251.2), and BIC (5456.49), indicating better model fit compared to the preceding models (Table 4).

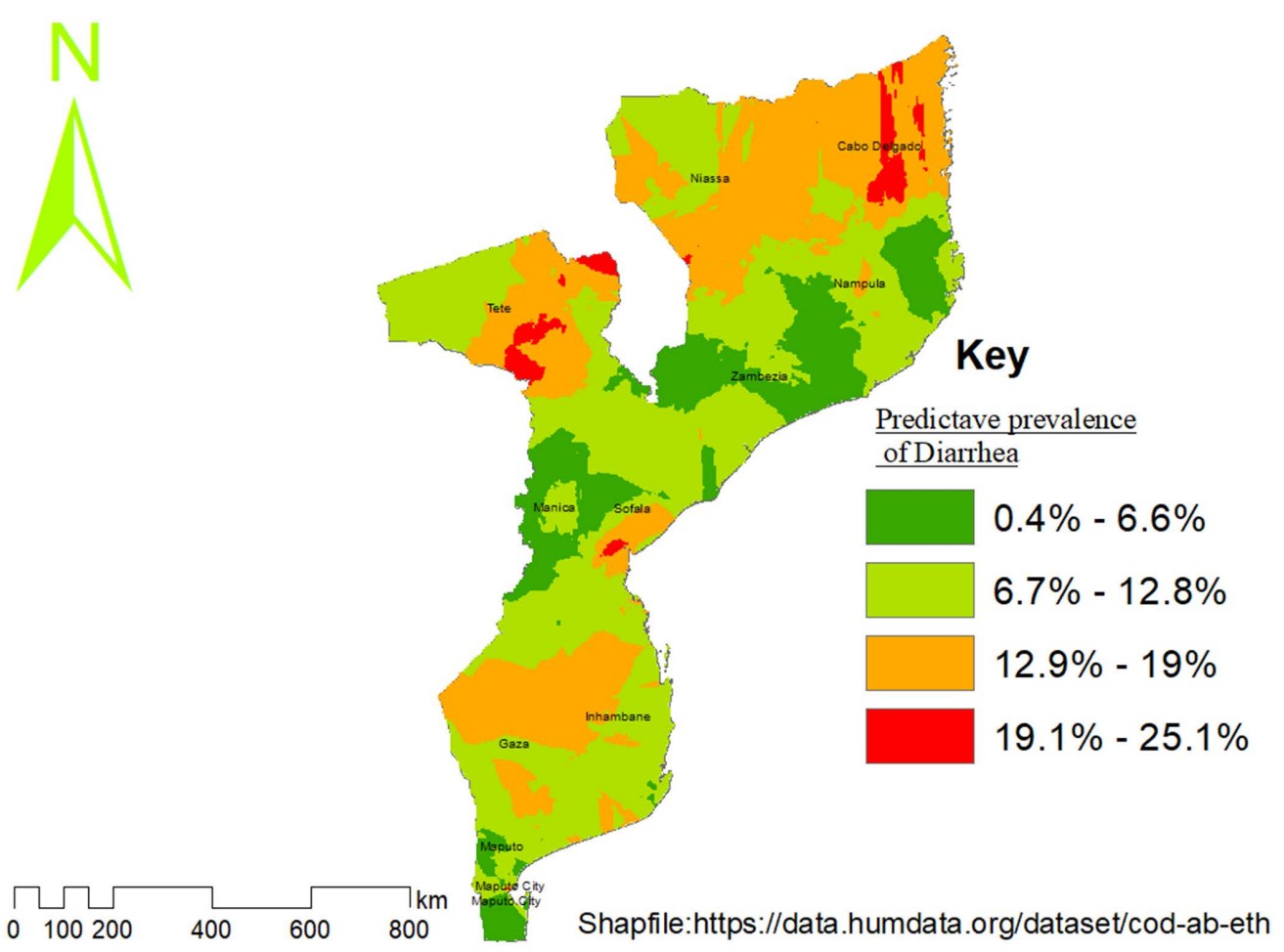

**Fig 4. Predicted prevalence of childhood diarrhea using ordinary Kriging, Mozambique DHS 2022/2023.**

## Discussion

Based on the 2022/2023 DHS data, the national prevalence of diarrhea among children in Mozambique was 8.8% (95% CI: 7.8–9.6), representing a modest decline from 10.5% in 2011 and 14.2% in 2003 [10], indicating gradual improvements in child health and sanitation over the past two decades. Diarrhea remains a leading cause of under-five mortality in low- and middle-income countries, particularly when prevalence exceeds 5% without effective intervention [38]. Mozambique's 8.8% prevalence indicates a moderate public health burden, although it is lower than in some sub-Saharan countries [29,39]. This underscores the need for multisectoral strategies to reduce incidence and improve treatment. As part of SDG 3, WHO and UNICEF promote integrated interventions: breastfeeding, rotavirus vaccination, zinc, ORT, and WASH [39,40]. While Mozambique shows progress, targeted efforts are still needed in high-burden districts.

The 2022/2023 DHS analysis shows that regional disparities in under-five diarrhea prevalence across Mozambique, consistent patterns are seen in Uganda and Malawi, where rural-urban inequality, poverty, and low education contribute to high rates [41]. Niassa (14.6%) and Cabo Delgado (14.1%) had the highest prevalence. It might be linked to poor health infrastructure, insecurity, and inadequate sanitation challenges, worsened by ongoing conflict in northern regions [42,43].

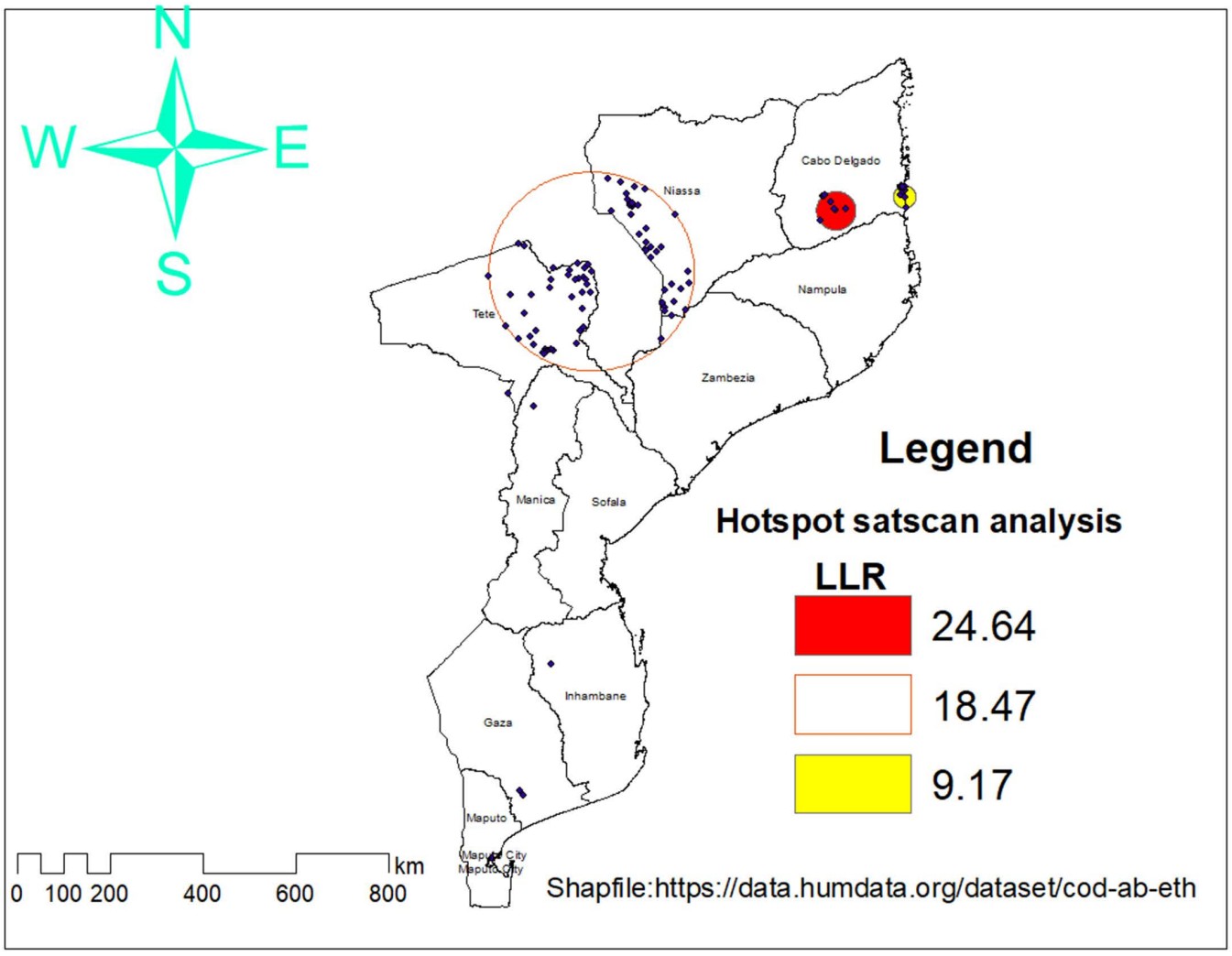

**Fig 5. Significant spatial clusters of childhood diarrhea identified by SaTScan, Mozambique DHS 2022/2023.**

**Table 3. Statistically significant spatial clusters of diarrhea among under-five children, Mozambique DHS 2022/2023.**

| Cluster | Location IDs Included | Coordinates/ Radius | Population | Number of Cases | Relative Risk | LL Ratio | P-value |
|---|---|---|---|---|---|---|---|
| 1 | 106, 107, 86, 95, 100, 65, 69, 68, 67, 66 | (13.419523 S, 39.218228 E) / 41.38 km | 196 | 56 | 2.79 | 24.64 | <0.001 |
| 2 | 266, 264, 265, 258, 298, 299, 263, 272, 269, 267, 300, 271, 301, 268, 270, 302, 284, 283, 294, 30, 31, 13, 49, 50, 293, 285, 28, 16, 29, 26, 51, 278, 32, 292, 43, 15, 279, 5, 7, 6, 1, 4, 281, 2, 3, 45, 282, 39, 17, 25, 40, 277, 54, 41, 261, 260, 291, 44, 254, 247, 252, 253, 255, 307, 20, 251, 23, 249, 256, 250, 24, 53, 257, 220, 8, 48, 306, 27, 42, 259, 280 | (14.593186 S, 34.490840 E) / 213.63 km | 1,292 | 203 | 1.61 | 18.46 | <0.001 |

**Table 4. Multilevel logistic regression models for individual- and community-level determinants of diarrhea among under-five children in Mozambique, DHS 2022/2023.**

| Variables | Model 0 | Model 1 (AOR, 95% CI) | Model 2 (AOR, 95% CI) | Model 3 (AOR, 95% CI) |
|---|---|---|---|---|
| **Child age (0–11 ref)** | | | | |
| 12–23 months | | 1.35 (1.11–1.66) | | 1.36 (1.11–1.66) * |
| 24–59 months | | 0.498 (0.40–0.58) | | 0.49 (0.41–0.59) * |
| **Maternal education (Secondary+ ref)** | | | | |
| Uneducated | | 0.82 (0.63–1.05) | | 0.80 (0.62–1.03) |
| Primary | | 0.76 (0.60–0.93) | | 0.77 (0.63–0.96) * |
| **Wealth index (Medium ref)** | | | | |
| Low | | 0.98 (0.79–1.22) | | 0.96 (0.77–1.19) |
| High | | 0.02 (0.80–1.30) | | 0.95 (0.71–1.26) |
| **Place of residence (Urban ref)** | | | | |
| Rural | | | 0.69 (0.52–0.91) | 0.69 (0.52–0.91) * |
| **Region (Niassa ref)** | | | | |
| Cabo Delgado | | | 1.06 (0.75–1.51) | 0.76 (0.54–1.07) |
| Nampula | | | 0.42 (0.29–0.62) | 0.42 (0.29–0.61) * |
| Zambézia | | | 0.36 (0.23–0.56) | 0.35 (0.23–0.55) * |
| Tete | | | 0.95 (0.66–1.37) | 0.95 (0.66–1.37) |
| Manica | | | 0.33 (0.22–0.51) | 0.30 (0.20–0.45) * |
| Sofala | | | 0.63 (0.43–0.93) | 0.51 (0.34–0.75) * |
| Inhambane | | | 0.71 (0.46–1.10) | 0.49 (0.32–0.76) * |
| Gaza | | | 0.77 (0.50–1.18) | 0.54 (0.35–0.82) * |
| Maputo Province | | | 0.30 (0.18–0.51) | 0.26 (0.15–0.44) * |
| Maputo City | | | 0.67 (0.42–1.06) | 0.50 (0.32–0.80) * |
| **Random Effects** | | | | |
| Cluster-level variance | 0.59 | 0.48 | 0.39 | 0.30 |
| ICC (%) | 15.2 | 12.8 | 10.7 | 8.5 |
| MOR | 2.08 | 1.94 | 1.81 | 1.69 |
| PCV (%) | Reference | 18.6 | 33.9 | 49.2 |
| **Model Comparison** | | | | |
| Deviance (−2LLR) | 2887.6 | 2638.4 | 2840.9 | 2596.5 |
| AIC | 5779.3 | 5310.9 | 5709.8 | 5251.2 |
| BIC | 5793.5 | 5431.3 | 5808.9 | 5456.49 |

**Note:** AOR = Adjusted Odds Ratio; CI = Confidence Interval; ICC = Intra-class Correlation Coefficient; MOR = Median Odds Ratio; PCV = Proportional Change in Variance. $p < 0.05$.

Surprisingly, Maputo City also reported a high rate (13.4%), likely due to overcrowded informal settlements with poor sanitation and limited piped water access [44]. As in other urban areas across sub-Saharan Africa, urban poverty remains a key driver of diarrheal disease through overcrowding and unsafe hygiene conditions [45].

In contrast, Maputo Province recorded the lowest prevalence (5.2%). The province may have better infrastructure and health service coverage compared to other provinces. The region's high affluence and substantial health system investment could potentially enhance child health outcomes [46]. Similarly, the low diarrhea prevalence in Zambézia (5.5%), Manica (5.9%), and Nampula (6.5%) is unexpected given their historically high burden. This may reflect recent gains in

WASH efforts, expanded community outreach, or seasonal factors that temporarily lowered transmission during the survey period [47].

Provinces with moderate prevalence, such as Tete (12.3%), Gaza (12.0%), Inhambane (11.1%), and Sofala (10.4%), present a mixed picture. The regions in question are subjected to varying degrees of climatic stress, such as floods and droughts, which significantly impact their water safety and hygiene practices [48]. The study reveals persistent environmental and behavioral risk factors in childhood diarrhea in Mozambique, highlighting the need for tailored interventions in northern regions (Niassa and Cabo Delgado) and lower-prevalence provinces [43].

The prevalence of diarrhea among children in urban areas is 11.7% (95% CI: 9.97%–13.36%), while it is 7.5% in rural areas (95% CI: 6.56%–8.66%). Urban children in Mozambique face a higher diarrhea risk than rural peers, likely due to poor sanitation, unsafe water, and inadequate waste management in informal settlements [29]. These conditions increase environmental contamination and diarrheal pathogen transmission. Overcrowding in urban slums further amplifies fecal-oral spread due to close contact and poor hygiene facilities [49]. Urban areas face increased diarrhea risk due to public sanitation issues, food contamination, and poor storage, with urban caregivers reporting more cases, especially in sub-Saharan Africa [50,51]. The analysis highlights Mozambique's urban vulnerability, particularly among children, and calls for urgent urban-targeted WASH interventions like safe water access, improved sanitation, and health education. UNICEF's 2019 analysis reveals urban slums often have worse sanitation than rural villages, despite rural children's 31% lower diarrhea prevalence [52]. Certain regions, including Maputo Province, Manica, and Zambézia, may lower diarrhea rates in children due to improved health programs and WASH coverage, supported by international NGOs[30]. The current findings in Mozambique align with regional patterns across Africa, showing diarrhea is more prevalent in conflict-affected, poor, and infrastructure-deficient areas, both rural and urban [51]. The SaTScan analysis revealed statistically significant high-risk clusters of diarrhea cases. The most likely cluster (Cluster 1), with a relative risk (RR) of 2.79, contains the northern districts of Niassa and Cabo Delgado. The second most likely cluster (Cluster 2), with an RR of 1.61, covers central Mozambique. The study identifies clusters of high diarrhea prevalence in regions with limited access to clean water and sanitation, supporting targeted intervention in these areas, similar to Ethiopia and Nigeria [6,53].

Children aged 12–23 months had significantly higher odds of experiencing diarrhea (AOR = 1.36; 95% CI: 1.11–1.66) compared to those under one year, while children aged 24–59 months were significantly less likely to have diarrhea (AOR = 0.49; 95% CI: 0.41–0.59). Previous studies, have identified a vulnerability linked to weaning practices, increased mobility, and exposure [54–56]. Similar trends were reported in studies from Ethiopia and Nigeria [57,58], and a pooled analysis across SSA found diarrhea incidence peaking between 12 and 23 months [59]. This study is in line with studies conducted at Tanzania and Ethiopia showed a lower likelihood of developing diarrhea, possibly due to immunity from earlier exposures [55,56,60,61].

Children of employed mothers have 1.2 times higher diarrhea odds due to reduced childcare time and reliance on alternate caregivers, with maternal employment also linked to increased risk of childhood illness [62]. This may be due to reduced time for child care, feeding, hygiene, and health-seeking practices when mothers are engaged in outside work [63–65]. Evidence from Nigeria and Ethiopia supports reduced breastfeeding and increased environmental risks [3,66], possibly due to time constraints in low-income settings. Children of mothers with primary education had significantly lower odds of diarrhea (AOR = 0.77; 95% CI: 0.63–0.96) compared to those whose mothers had secondary education or higher. Basic educational attainment significantly improves child health outcomes, including nutrition, healthcare utilization, hygiene practices, and child diarrhea, despite a seemingly small odds reduction in East Africa, like Mozambique and Ghana [67,68]. A higher health-seeking behavior index is linked to nearly 5 times higher diarrhea rates, possibly due to mothers actively seeking healthcare. A similar phenomenon was observed in Kenya [69]. This may reflect reporting bias rather than actual disease burden. The observed higher prevalence among children of educated mothers may reflect reporting bias, as more educated mothers may be more likely to recognize and report diarrheal symptoms. Alternatively, this finding may not indicate a true increased risk and should be interpreted with caution.

The study found that children in Maputo Province had a 74% lower odds of diarrhea, possibly due to improved infrastructure and health services, consistent with study in southern Mozambique [70]. Maputo City demonstrated a decrease in odds (AOR = 0.50), indicating improved WASH access and vaccine coverage, as reported by UNICEF [29]. Zambézia and Manica showed significantly lower odds (AOR = 0.35 and 0.30) of developing a disease, possibly due to the effectiveness of hygiene campaigns and maternal health programs [29]. Inhambane and Gaza experienced reduced odds (AOR = 0.49 and 0.54) of greater access to improved sanitation, similar to central-southern Mozambique, where access to improved toilets was higher [70].

Children living in several regions of Mozambique had significantly lower chances of having diarrhea compared to those in Niassa. Specifically, the risk was lower in Nampula (58% lower, AOR = 0.42), Zambézia (65% lower, AOR = 0.35), Manica (70% lower, AOR = 0.30), Sofala (49% lower, AOR = 0.51), Inhambane (51% lower, AOR = 0.49), Gaza (46% lower, AOR = 0.54), Maputo Province (74% lower, AOR = 0.26), and Maputo City (50% lower, AOR = 0.5. The study indicates disparities in environmental and healthcare conditions across regions, possibly influenced by factors such as water safety, infrastructure, climate, and localized interventions. Studies from Kenya, Tanzania, and other parts of Mozambique revealed comparable spatial disparities [8,10].. The study found that significant variation in diarrhea risk was due to cluster-level differences, with an ICC decline from 15.2% to 8.5%, MOR decreasing from 2.08 to 1.69, and PCV reaching 49.2%. The model's variables were found to explain nearly half of the variation between clusters. Similar ICC and MOR estimates were found in multilevel studies conducted in Ethiopia and Nigeria [71,72].

The 2022/2023 Mozambique DHS provides valuable insights through its nationally representative design, standardized questionnaires, and integration of geospatial and biomarker data, allowing robust statistical power, precise estimates, and detailed subgroup analyses. Its recency and contextual depth make it a strong foundation for evidence-based interventions. However, the cross-sectional design limits causal inference, maternal recall may introduce reporting bias, GPS displacement and spatial aggregation may cause ecological fallacies, and the lack of pathogen-specific and seasonal data may affect precision. Despite these limitations, targeting high-burden regions and addressing determinants such as maternal education, rural access, and health-seeking behavior aligns with global strategies to reduce under-five mortality and achieve SDG 3 [23,73]. Evidence from sub-Saharan Africa indicates that community- and household-level interventions can significantly reduce diarrhea-related morbidity, emphasizing the importance of context-specific approaches [74,75].

## Conclusions

Diarrhea continues to pose a moderate public health challenge for children in Mozambique, with clear disparities across regions. Factors such as child age, maternal education, rural versus urban residence, and geographic location influence the risk of childhood diarrhea. Future interventions should focus on age- and area-specific risk factors. Targeted interventions include expanding maternal education, improving WASH infrastructure, and enhancing maternal support systems, including family support, access to healthcare services, and community health programs, particularly in high-burden regions such as Niassa and Cabo Delgado. Implementing these measures can help reduce childhood diarrhea and support Mozambique's progress toward child health and Sustainable Development Goal 3.

## Acknowledgments

We would like to thank the Mozambique National Institute of Statistics (INE) for providing access to the dataset used in this study, and the DHS Program for granting permission to download the necessary supporting materials.

## Author contributions

**Conceptualization:** Thomas Kidanemariam Yewodiaw, Mihret Getnet, Mequanent Dessie Bitewa.

**Data curation:** Thomas Kidanemariam Yewodiaw, Mihret Getnet, Hiwot Tezera Endale.

**Formal analysis:** Thomas Kidanemariam Yewodiaw, Mihret Getnet, Mequanent Dessie Bitewa, Hiwot Tezera Endale.

**Funding acquisition:** Mihret Getnet.

**Investigation:** Thomas Kidanemariam Yewodiaw, Mequanent Dessie Bitewa.

**Methodology:** Thomas Kidanemariam Yewodiaw, Hiwot Tezera Endale.

**Project administration:** Hiwot Tezera Endale.

**Software:** Mequanent Dessie Bitewa, Hiwot Tezera Endale.

**Supervision:** Mequanent Dessie Bitewa, Hiwot Tezera Endale.

**Validation:** Thomas Kidanemariam Yewodiaw.

**Visualization:** Mihret Getnet, Mequanent Dessie Bitewa.

**Writing – original draft:** Thomas Kidanemariam Yewodiaw.

**Writing – review & editing:** Thomas Kidanemariam Yewodiaw, Mihret Getnet, Mequanent Dessie Bitewa, Hiwot Tezera Endale.

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
