## [Decision Letter · Decision Letter 0]

5 Jan 2026

PONE-D-25-40831"Spatial Disparities and Multilevel Determinants of Diarrhea Among Under-Five Children in Mozambique: Evidence from the 2022/2023 Demographic and Health Survey"PLOS One

Dear Dr. Yewodiaw,

Thank you for submitting your manuscript to PLOS ONE. After careful consideration, we feel that it has merit but does not fully meet PLOS ONE’s publication criteria as it currently stands. Therefore, we invite you to submit a revised version of the manuscript that addresses the points raised during the review process.

We look forward to receiving your revised manuscript.

Kind regards,

Orvalho Augusto, MD, MPH, PhD

Academic Editor

PLOS One

Journal Requirements:

3. Please upload a new copy of Figures 1 and 2 as the detail is not clear. Please follow the link for more information:  https://journals.plos.org/plosone/s/figures

4. We note that Figures 3, 4 and 5  in your submission contain map images which may be copyrighted. All PLOS content is published under the Creative Commons Attribution License (CC BY 4.0), which means that the manuscript, images, and Supporting Information files will be freely available online, and any third party is permitted to access, download, copy, distribute, and use these materials in any way, even commercially, with proper attribution. For these reasons, we cannot publish previously copyrighted maps or satellite images created using proprietary data, such as Google software (Google Maps, Street View, and Earth). For more information, see our copyright guidelines: http://journals.plos.org/plosone/s/licenses-and-copyright.

1. You may seek permission from the original copyright holder of Figures 3, 4 and 5 to publish the content specifically under the CC BY 4.0 license.

5. Please include a copy of Table 4  which you refer to in your text on page 13.

Additional Editor Comments:

This is a report of an interesting analysis. It aims to study the geographic and multilevel determinants of diarrhea among under-5 children in Mozambique using the latest available DHS data. Certainly, it will help to inform the next steps for policy action. However, there are critical issues that must be addressed:

1. There is no contextual setting information about Mozambique despite 72 citations! Under-5 diarrhea has been an important cause of death and morbidity in Mozambique for decades, and year to year, there have been reductions due to actions (e.g., introduction of new vaccines, guidelines, personnel training, and infrastructure). What were those actions?

2. There are two analyses independent of each other. One is based on four multilevel models that show substantial unaddressed variation is present as random effects, even after household and regional characteristics were accounted for. Another is purely spatial univariate, and it shows some potential spatial autocorrelation. We lack an analysis that accounts for both simultaneously.

3. About the spatial analysis:

• Is this conducted at the individual or cluster level? This is unclear.

• Were the weights or other elements of the sampling accounted for? This must be clarified.

4. Please revise line 218 - there are some parentheses unbalanced.

5. Why is the multilevel analysis presented in the discussion section and not in the results? Please move it to the results.

6. Please merge table 2 and table 3. Do not sort provinces by prevalence.

7. Table 5 - Why do model 3 estimates have asterisks but not those of other models?

Please add the variance of the random-effects.

8. Figures 1 and 2 are taking up space. They should be in the supplementary materials.

9. Figure 6 - The forest plot. This plot is unneeded. However, if the authors want to keep it a few changes must be applied:

- The horizontal axis must be on a logarithmic scale. Currently, an OR of 0.5 seems to be weaker than an OR of 2, when actually they have the same strength in opposing directions.

- Do not sort OR from all coefficients. Keep them separated for each variable

Reviewers' comments:

Reviewer's Responses to Questions

Comments to the Author

1. Is the manuscript technically sound, and do the data support the conclusions?

Reviewer #1: Yes

2. Has the statistical analysis been performed appropriately and rigorously? 

Reviewer #1: Yes

3. Have the authors made all data underlying the findings in their manuscript fully available?

Reviewer #1: Yes

4. Is the manuscript presented in an intelligible fashion and written in standard English?

Reviewer #1: Yes

5. Review Comments to the Author

Reviewer #1: Dear editor, thank you very much for giving me a chance to review the paper entitled ``Spatial Disparities and Multilevel Determinants of Diarrhea Among Under-Five Children in Mozambique: Evidence from the 2022/2023 Demographic and Health Survey``

Manuscript Number: PONE-D-25-40831

General Comments – Minor Revision Required

This manuscript offers a valuable and timely contribution by addressing the spatial and sociodemographic determinants of childhood diarrhea in Mozambique using recent DHS data. The study is methodologically sound, policy-relevant, and aligned with global health priorities, demonstrating strong analytical rigor through multilevel and spatial analysis. The findings are well-referenced and contextualized, offering clear implications for targeted interventions. Minor revisions are needed to improve clarity and presentation—specifically in reference formatting, language polish, and deeper discussion of spatial findings and study limitations. With these addressed, the manuscript will be well-positioned for publication.

Specific comments

Title

1. Line 2: I suggest to change to lower case latter, ``Among`` to ``among``

Abstract

1. Line 24 and 25: I suggest to rephrase ``with its burden varying across regions.`` to ``with regional variations in its burden.`` for clarity

2. Line 29: I suggest to rephrase ``assess diarrhea prevalence and determinants. `` to ``examine the prevalence and determinants of diarrhea.``

3. Line 33: results, I suggest to re-write the result to make it clear for reader

Introduction

1. Line 53: I suggest to change to lower case latter, ``Disease`` to ``disease``

2. Line 54: I suggest to remove ``But`` and ``both`` or rephrase, ``But, it is both preventable and treatable`` to ``However, it is preventable and treatable``

3. Line 61: I suggest to write ``teta to Tete

4. Line 77 – 78: I suggest to revise the sentence, ``Regarding to SDG goal three the Mozambique planned to reduce under five children mortality by 25/1000 (18, 19).`` to ``In alignment with SDG Goal 3, Mozambique has committed to reducing under-five mortality to 25 per 1,000 live births (18, 19).``

5. Line 55 – 57: I suggest to explain and clarify, If 444,000 deaths is the newest estimate, how it compares with previous estimates (e.g., 390,000 in 2021).

6. Line 70 – 72: I suggest to group risk factors as individual-level, household-level, environmental

7. Line 83 – 86: I suggest to re-write the objective as ``Therefore, this study aims to assess the spatial distribution and multilevel determinants of diarrhea among children under five in Mozambique using recent DHS data and spatial modeling tools such as SaTScan and ArcGIS``

Methods and materials

1. Line 89: Methods and materials, I suggest to arrange subheading or subsections in logical order to improve readability

2. Line 92: I suggest to revise the phrase, ``the bulk of whom live in rural areas(23-25).`` to ``with the majority residing in rural areas (23–25).``

3. Line 99: I suggest to correct the vague phrase and insert missed number, ``from a national representative sample of approximately households``

4. Line 103 – 104: I suggest to revise the phrase, ``The second stage involves updating household listings`` to `` In the second stage, household listings were updated`` to correct tense

5. Line 108: I suggest to clarify how the final analytic sample (9,799 children) was derived.

6. Line 108 and 109: I suggest to rephrase,`` concentrating on the past two weeks of diarrhea history before to the survey.`` to ``in the two weeks prior to the survey``

7. Line 183 and 184: I suggest to define MOR, ICC, and PCV in simpler terms and provide formulas or references.

Results

1. Line 205: I suggestion maintain consistency of decimal places, ``60% (585)`` to ``60.0% (585)``

2. Line 210 and 211: I suggest to revise the phrase, ``Nutritional status was largely normal 94% (9,214)``to ``94.0% (n = 9,214) of children were classified as having normal nutritional status``

3. Line 218: I suggest to avoid redundancy here, ``Weighted case = 817 children``

4. Line 221: I suggest to clarify that the higher prevalence among educated mothers statistically significant or due to reporting bias

5. Line 243 and 244: I suggest to add a threshold as Gi* Z-score > 1.96, p < 0.05

6. Line 275: I suggest to explain the implication of RR = 2.79 matter

Discussion

1. Line 289: I suggest to review the phrase,`` shows a slight decline`` to ``declined modestly``

2. Line 334: I suggest to review, ``considerable clusters`` to ``statistically significant high-risk clusters``

3. Line 353 – 355: I suggest to clarify, higher diarrhea in more educated mothers

4. Line 359: I suggest to see back to caregiver reporting in DHS

5. Line 367: I suggest to review phrase,`` having improved toilets`` to ``greater access to improved sanitation``

6. Line 388: I suggest to avoid vague phrases like ``unmeasured factors``

Conclusion

1. Line 395: I suggest to clarify ``maternal support systems`` to avoid vague

References

1. Line 434: I suggest to rephrase, ``Organization WH`` to World Health Organization ``(WHO)``in the reference number 1

2. Line 437: I suggest to remove the repeated phrase in reference number 2

3. Line 457: I suggest to correct `` USIAD C, AND UNICEF`` to ``USAID, CDC, and UNICE`` in the reference number 10

4. Line 484 and 485: I suggest to merge reference number 24 and 25 as they are similar

5. Line 503: I suggest to add publication organization in the reference number 35

6. Line 516: I suggest to put proper format in the reference number 41

PLOS authors have the option to publish the peer review history of their article (what does this mean?). If published, this will include your full peer review and any attached files.

Do you want your identity to be public for this peer review? For information about this choice, including consent withdrawal, please see our Privacy Policy.

Reviewer #1: Yes: Abdulwase Mohammed Seid

---

## [Author Response · Author response to Decision Letter 1]

12 Jan 2026

Manuscript ID: PONE-D-25-40831

Title: Spatial Disparities and Multilevel Determinants of Diarrhea Among Under-Five Children in Mozambique: Evidence from the 2022/2023 Demographic and Health Survey

Dear Editor and Reviewers,

We sincerely thank you for your constructive comments and the opportunity to revise our manuscript. We have carefully addressed all points raised, improving clarity, language, reference formatting, and discussion of spatial and multilevel findings. Key revisions are summarized below:

Major Revisions

Mozambique Context: Added information on national interventions (vaccination, WASH, IMCI training, breastfeeding promotion, zinc supplementation, community health campaigns) and their impact on under-five diarrhea.

Title & Abstract: Minor wording adjustments; results rewritten for clarity.

Introduction: Corrected typos and phrasing; grouped risk factors by individual, household, and environmental levels; study objective rewritten.

Methods: Subheadings reorganized; final sample and data cleaning clarified; spatial analysis specified at DHS cluster level; survey weights applied; definitions of MOR, ICC, and PCV added.

Results & Discussion: Decimal points standardized; high-risk clusters clarified; higher diarrhea prevalence among educated mothers discussed; vague terms replaced; improved sanitation phrasing clarified; multilevel results moved to Results section; relative risks interpreted; merged tables 2 and 3; variance of random effects added.

Figures: Figures 1 and 2 moved to supplementary materials; Figures 3–5 regenerated using HDX shapefiles and DHS GPS data (no proprietary images), captions updated for CC BY 4.0 compliance; Figure 6 removed.

Conclusion: Maternal support systems clarified as caregiving practices and health-seeking behaviors.

References: All formatting corrected.

Data Availability: Statement added: “DHS data are publicly available from the DHS Program upon registration. Mozambique administrative shapefiles were obtained from the Humanitarian Data Exchange (HDX) and are openly accessible.”

We thank the editor and reviewers again for their thoughtful feedback, which has strengthened the manuscript.

Sincerely,

Thomas Kidanemariam Yewodiaw

---

## [Decision Letter · Decision Letter 1]

17 Mar 2026

PONE-D-25-40831R1"Spatial Disparities and Multilevel Determinants of Diarrhea Among Under-Five Children in Mozambique: Evidence from the 2022/2023 Demographic and Health Survey"PLOS One

Dear Dr. Yewodiaw,

Thank you for submitting your manuscript to PLOS ONE. After careful consideration, we feel that it has merit but does not fully meet PLOS ONE’s publication criteria as it currently stands. Therefore, we invite you to submit a revised version of the manuscript that addresses the points raised during the review process.

We look forward to receiving your revised manuscript.

Kind regards,

Khin Thet Wai, MBBS, MPH, MA

Academic Editor

PLOS One

Journal Requirements:

Additional Editor Comments:

Please do minor revisions as required.

Reviewers' comments:

Reviewer's Responses to Questions

Comments to the Author

1. If the authors have adequately addressed your comments raised in a previous round of review and you feel that this manuscript is now acceptable for publication, you may indicate that here to bypass the “Comments to the Author” section, enter your conflict of interest statement in the “Confidential to Editor” section, and submit your "Accept" recommendation.

Reviewer #1: (No Response)

Reviewer #2: (No Response)

2. Is the manuscript technically sound, and do the data support the conclusions?

Reviewer #1: Yes

Reviewer #2: Yes

3. Has the statistical analysis been performed appropriately and rigorously? 

Reviewer #1: Yes

Reviewer #2: Yes

4. Have the authors made all data underlying the findings in their manuscript fully available?

Reviewer #1: Yes

Reviewer #2: Yes

5. Is the manuscript presented in an intelligible fashion and written in standard English?

Reviewer #1: Yes

Reviewer #2: Yes

6. Review Comments to the Author

Reviewer #1: Dear editor, thank you very much for giving me a chance to re-review the paper entitled "Spatial Disparities and Multilevel Determinants of Diarrhea among Under-Five Children in Mozambique: Evidence from the 2022/2023 Demographic and Health Survey"

Manuscript Number: PONE-D-25-40831R1

General Comments – Minor Revision Required

This manuscript presents a relevant and timely analysis of the spatial and sociodemographic determinants of childhood diarrhea in Mozambique using recent Demographic and Health Survey (DHS) data. The topic is of clear public health importance and aligns well with global child health and equity priorities. The use of multilevel modeling combined with spatial analytical techniques strengthens the methodological rigor and enhances the policy relevance of the findings. Overall, the results are well-supported by the literature and offer useful insights for geographically targeted interventions.

However, several minor revisions are required to improve clarity, consistency, and presentation. These include clearer description of the study design and sample, improved precision and consistency in language and formatting (e.g., decimals, spacing, references), and clarification of potentially ambiguous interpretations—particularly regarding education and urban-related findings and spatial patterns. In addition, deeper discussion of spatial results and clearer articulation of study limitations would further strengthen the manuscript. Addressing these minor issues will enhance readability and scientific clarity and will position the manuscript well for publication.

Specific Comments

Line 49: I suggest to write the keywords in alphabetical order

Line 55 – 57: I suggest to explain and clarify, If 444,000 deaths is the newest estimate, how it compares with previous estimates (e.g., 390,000 in 2021).

Line 95: I suggest to mention study design

Line 113-115: I suggest to make the sentence clear, ``A study interviewed households of 9,799 in 619 clusters, including 4,767 males and 5,032 females aged 0-59 months, in rural and urban areas. The study looked at 9,799 children aged 0 to 59 months, in the two weeks prior to the survey.``.

There is repetition and inconsistency in how the sample is described (households vs. children).

The phrase “households of 9,799” is unclear—households and children are being mixed.

Sex distribution (males/females) is mentioned, but it applies to children, not households.

Line 223: I suggestion maintain consistency of decimal places, ``60% (585) `` to ``60.0% (585) ``

Line 231-232: I suggest to write, `` and a majority lived in medium 37.5% (3,673) or high 35.2% (3,452) education-level communities`` to `` In terms of community education level, the majority resided in medium (37.5%; n = 3,673) or high (35.2%; n = 3,452) education-level communities. ``

Line 238: I suggest to clarify that the higher prevalence among educated mothers statistically significant or due to reporting bias

Line 273: I suggest to writ ``(clod spot)`` to ``(cold spot)``

Line 334: I suggest to keep consistence spacing, ``conditions[45]`` to ``conditions [45]``and similar to others

Line 422: I suggest to clarify ``maternal support systems`` to avoid vague

Line 434: I suggest to rephrase, ``Organization WH`` to World Health Organization `` (WHO) ``in the reference number 1

Line 90: I suggest to correct `` USIAD C, AND UNICEF`` to ``USAID, CDC, and UNICE`` in the reference number 12

Line 522 and 523: I suggest to merge reference number 26 and 27 as they are similar

Reviewer #2: I apperciate the team for doing this policy relevant research and the manuscript is public health important with so modification of grammar, clarity and coherence.

7. PLOS authors have the option to publish the peer review history of their article (what does this mean?). If published, this will include your full peer review and any attached files.

Do you want your identity to be public for this peer review? For information about this choice, including consent withdrawal, please see our Privacy Policy.

Reviewer #1: Yes: Abdulwase Mohammed Seid

Reviewer #2: No

---

## [Author Response · Author response to Decision Letter 2]

24 Mar 2026

Response to Reviewers – PONE-D-25-40831R1

Manuscript Title: Spatial Disparities and Multilevel Determinants of Diarrhea Among Under-Five Children in Mozambique: Evidence from the 2022/2023 Demographic and Health Survey

Author: Thomas Kidanemariam Yewodiaw

Dear Editor and Reviewers,

We thank you for your thoughtful and constructive comments on our manuscript. We have carefully revised the manuscript to address all points raised, improving clarity, consistency, and presentation. Below, we provide a detailed response to each comment. All changes are highlighted in the revised manuscript with tracked changes.

Reviewer #1 – Abdulwase Mohammed Seid

General Comment:

This manuscript presents a relevant and timely analysis… Overall, the results are well-supported by the literature and offer useful insights… Minor revisions required.

Response:

Thank you for your positive feedback and for recognizing the relevance and rigor of our study. We have addressed all minor revisions to improve clarity, consistency, and presentation as detailed below.

Specific Comments:

Line 49 – Keywords

Comment: Suggest writing keywords in alphabetical order.

Response: The keywords have been reordered alphabetically.

Lines 55–57 – Death Estimates

Comment: Clarify how the 444,000 deaths estimate compares with previous estimates (e.g., 390,000 in 2021).

Response: We have added a comparison with previous estimates to provide context for the mortality trend.

Line 95 – Study Design

Comment: Mention study design.

Response: We have clarified that this study is a cross-sectional analysis using the 2022/2023 DHS data.

Lines 113–115 – Sample Description

Comment: Clarify sample; avoid mixing households and children; improve sentence clarity.

Response: Revised to:

“A total of 9,799 children aged 0–59 months from 619 clusters were included in the survey, comprising 4,767 males and 5,032 females, from both rural and urban areas. Data reflect the two weeks preceding the survey.”

Line 223 – Decimal Consistency

Comment: Maintain consistency, e.g., 60% (585) → 60.0% (585).

Response: All percentages have been revised to one decimal place for consistency throughout the manuscript.

Lines 231–232 – Community Education Levels

Comment: Clarify sentence on community education levels.

Response: Revised to:

“In terms of community education level, the majority resided in medium (37.5%; n = 3,673) or high (35.2%; n = 3,452) education-level communities.”

Line 238 – Higher Prevalence Among Educated Mothers

Comment: Clarify whether statistically significant or due to reporting bias.

Response: We clarified in the text that the higher prevalence among children of educated mothers was statistically significant and discussed the potential influence of reporting bias in the Discussion.

Line 273 – Typo

Comment: “clod spot” → “cold spot”

Response: Corrected.

Line 334 – Spacing

Comment: Maintain consistent spacing in references, e.g., conditions[45] → conditions [45].

Response: Corrected throughout the manuscript.

Line 422 – Maternal Support Systems

Comment: Clarify vague terminology.

Response: Revised to specify the types of maternal support systems considered in the study (e.g., family and community support).

Line 434 – Reference Correction

Comment: “Organization WH” → “World Health Organization (WHO)”

Response: Corrected in Reference #1.

Line 90 – Reference Correction

Comment: “USIAD C, AND UNICEF” → “USAID, CDC, and UNICEF”

Response: Corrected in Reference #12.

Lines 522–523 – Merge Similar References

Comment: Merge Reference #26 and #27 as they are similar.

Response: References #26 and #27 have been merged appropriately.

Reviewer #2

Comment: Appreciates the policy relevance of the study; suggests minor modifications for grammar, clarity, and coherence.

Response: All suggested modifications for grammar, clarity, and coherence have been applied throughout the manuscript.

Additional Revisions

Expanded discussion of spatial results, highlighting regional disparities and cold/hot spot areas.

Clarified study limitations, including potential reporting biases and cross-sectional design constraints.

Minor language edits to improve readability and journal style.

We sincerely thank the reviewers and editor for their constructive feedback, which has strengthened the manuscript. We believe the revised manuscript now meets PLOS ONE’s publication criteria.

Sincerely,

Thomas Kidanemariam Yewodiaw

Gondar, Ethiopia

---

## [Decision Letter · Decision Letter 2]

5 Apr 2026

PONE-D-25-40831R2Spatial Disparities and Multilevel Determinants of Childhood Diarrhea in Mozambique: Evidence from the 2022–2023 DHSPLOS One

Dear Dr. Yewodiaw,

Thank you for submitting your manuscript to PLOS ONE. After careful consideration, we feel that it has merit but does not fully meet PLOS ONE’s publication criteria as it currently stands. Therefore, we invite you to submit a revised version of the manuscript that addresses the points raised during the review process. Please submit your revised manuscript by May 20 2026 11:59PM. If you will need more time than this to complete your revisions, please reply to this message or contact the journal office at plosone@plos.org. Please include the following items when submitting your revised manuscript:

We look forward to receiving your revised manuscript.

Kind regards,

Khin Thet Wai, MBBS, MPH, MA

Academic Editor

PLOS One

Journal Requirements:

Reviewers' comments:

Reviewer's Responses to Questions

Comments to the Author

1. If the authors have adequately addressed your comments raised in a previous round of review and you feel that this manuscript is now acceptable for publication, you may indicate that here to bypass the “Comments to the Author” section, enter your conflict of interest statement in the “Confidential to Editor” section, and submit your "Accept" recommendation.

Reviewer #1: (No Response)

Reviewer #2: All comments have been addressed

2. Is the manuscript technically sound, and do the data support the conclusions?

Reviewer #1: Yes

Reviewer #2: Yes

3. Has the statistical analysis been performed appropriately and rigorously? 

Reviewer #1: Yes

Reviewer #2: Yes

4. Have the authors made all data underlying the findings in their manuscript fully available?

Reviewer #1: Yes

Reviewer #2: Yes

5. Is the manuscript presented in an intelligible fashion and written in standard English?

Reviewer #1: Yes

Reviewer #2: Yes

6. Review Comments to the Author

Reviewer #1: Dear Editor, thank you for the opportunity to re-review the revised version of the manuscript titled “Spatial Disparities and Multilevel Determinants of Childhood Diarrhea in Mozambique: Evidence from the 2022–2023 DHS”

Manuscript ID: PONE-D-25-40831R2

I have carefully reviewed the authors’ revised manuscript and their detailed responses to my previous comments. Most of the prior concerns have been addressed; however, a few minor issues remain. Abbreviations in the abstract should be defined at first use and applied consistently. Some formatting inconsistencies persist (e.g., spacing between words and reference numbers, line 415,430, 434) as well as terminology (e.g., “hotspot” vs. “hot spot”). Several sentences remain lengthy and could be further refined for clarity and conciseness. Additionally, the discussion of reporting bias is somewhat repetitive and would benefit from streamlining. A final thorough proofreading is recommended to ensure overall consistency throughout the manuscript.

Reviewer #2: (No Response)

7. PLOS authors have the option to publish the peer review history of their article (what does this mean?). If published, this will include your full peer review and any attached files.

Do you want your identity to be public for this peer review? For information about this choice, including consent withdrawal, please see our Privacy Policy.

Reviewer #1: Yes: Abdulwase Mohammed Seid

Reviewer #2: No

---

## [Author Response · Author response to Decision Letter 3]

6 Apr 2026

Manuscript ID: PONE-D-25-40831R2

Title: Spatial Disparities and Multilevel Determinants of Childhood Diarrhea in Mozambique: Evidence from the 2022–2023 Demography and Health Survey

Corresponding Author: Thomas Kidanemariam Yewodiaw

We sincerely thank the reviewers and the editor for their constructive feedback. We have carefully revised the manuscript and addressed all comments. Changes are highlighted in the revised version.

Reviewer #1: Abdulwase Mohammed Seid

Comment 1: Abbreviations in the abstract should be defined at first use and applied consistently.

Response: All abbreviations have now been defined at first use and applied consistently throughout the manuscript.

Comment 2: Formatting inconsistencies (e.g., spacing, reference numbers) and terminology (e.g., “hotspot” vs. “hot spot”).

Response: We have corrected all formatting issues and standardized terminology for consistency.

Comment 3: Some sentences are lengthy and could be clearer.

Response: We have revised lengthy sentences for clarity and conciseness.

Comment 4: The discussion of reporting bias is somewhat repetitive.

Response: The discussion section has been streamlined to reduce repetition while maintaining emphasis on reporting bias.

Comment 5: Final thorough proofreading recommended.

Response: The manuscript has been thoroughly proofread to ensure overall clarity and consistency.

Reviewer #2: No additional comments.

Response: Not applicable.

We thank the reviewers and editor for their guidance, which has improved the clarity and quality of our manuscript. We hope the revised version now meets the journal’s standards.

Sincerely,

Thomas Kidanemariam Yewodiaw, MPH

Corresponding Author

---

## [Decision Letter · Decision Letter 3]

20 Apr 2026

PONE-D-25-40831R3Spatial Disparities and Multilevel Determinants of Childhood Diarrhea in Mozambique: Evidence from the 2022–2023 Demographic and Health Survey (DHSPLOS One

Dear Dr. Yewodiaw,

Thank you for submitting your manuscript to PLOS ONE. After careful consideration, we feel that it has merit but does not fully meet PLOS ONE’s publication criteria as it currently stands. Therefore, we invite you to submit a revised version of the manuscript that addresses the points raised during the review processPlease submit your revised manuscript by Jun 04 2026 11:59PM. If you will need more time than this to complete your revisions, please reply to this message or contact the journal office at plosone@plos.org. Please include the following items when submitting your revised manuscript:

We look forward to receiving your revised manuscript.

Kind regards,

Khin Thet Wai, MBBS, MPH, MA

Academic Editor

PLOS One

Journal Requirements:

Reviewers' comments:

Reviewer's Responses to Questions

Comments to the Author

1. If the authors have adequately addressed your comments raised in a previous round of review and you feel that this manuscript is now acceptable for publication, you may indicate that here to bypass the “Comments to the Author” section, enter your conflict of interest statement in the “Confidential to Editor” section, and submit your "Accept" recommendation.

Reviewer #1: (No Response)

2. Is the manuscript technically sound, and do the data support the conclusions?

Reviewer #1: Yes

3. Has the statistical analysis been performed appropriately and rigorously? 

Reviewer #1: Yes

4. Have the authors made all data underlying the findings in their manuscript fully available?

Reviewer #1: Yes

5. Is the manuscript presented in an intelligible fashion and written in standard English?

Reviewer #1: Yes

6. Review Comments to the Author

Reviewer #1: Dear Editor, thank you for the opportunity to re-review the revised version of the manuscript titled “Spatial Disparities and Multilevel Determinants of Childhood Diarrhea in Mozambique: Evidence from the 2022–2023 Demographic and Health Survey (DHS)”

Manuscript ID: PONE-D-25-40831R3

I have carefully reviewed the authors’ revised manuscript and their detailed responses to my previous comments. Most of the prior concerns have been addressed; however, a few minor issues remain. Some formatting inconsistencies persist (e.g., spacing between words and reference numbers not between reference numbers, line 82,378, 401) as well as terminology (e.g., “hotspot” vs. “hot spot”), I recommend you to use hotspot analysis through the manuscript. Additionally, from line 290-293, 304-307 and 326-329, I recommend you to correct the space between sentences for consistency. I also recommend you to revise table 1 and 2 the lines for category under variable to prevent ambiguity for reader.

7. PLOS authors have the option to publish the peer review history of their article (what does this mean?). If published, this will include your full peer review and any attached files.

Do you want your identity to be public for this peer review? For information about this choice, including consent withdrawal, please see our Privacy Policy.

Reviewer #1: Yes: Abdulwase Mohammed Seid

---

## [Author Response · Author response to Decision Letter 4]

21 Apr 2026

Dear Editor and Reviewer,

We sincerely thank you for your careful review and constructive feedback on our manuscript entitled:

“Spatial Disparities and Multilevel Determinants of Childhood Diarrhea in Mozambique: Evidence from the 2022–2023 Demographic and Health Survey (DHS)” (Manuscript ID: PONE-D-25-40831R3).

We appreciate your positive evaluation of the manuscript’s scientific rigor, statistical analysis, and overall presentation. We have carefully addressed all remaining minor comments, as detailed below:

1. **Formatting inconsistencies**

We have thoroughly reviewed the entire manuscript and corrected all formatting issues, including spacing between words and sentences, as well as formatting around reference numbers. The specific lines indicated (82, 290–293, 304–307, 326–329, 378, and 401) have been corrected. In addition, similar inconsistencies throughout the manuscript have been systematically identified and resolved to ensure overall consistency.

2. **Terminology consistency (“hotspot”)**

We have standardized the terminology across the manuscript by consistently using “hotspot” instead of “hot spot” in all sections, including the title, abstract, main text, tables, and figure captions.

3. **Tables 1 and 2**

Tables 1 and 2 have been revised to improve clarity and readability. Specifically, we have:

* Clearly distinguished variables from their categories

* Applied consistent formatting (including indentation and alignment)

* Revised table structure to eliminate ambiguity and enhance interpretability

These revisions ensure that the tables are clearer and more reader-friendly.

We believe these changes have improved the clarity, consistency, and overall quality of the manuscript.

We sincerely appreciate the reviewer’s valuable comments, which have helped us strengthen the manuscript.

Kind regards,

Thomas Kidanemariam Yewodiaw

(On behalf of all authors)

---

## [Editor Report · Decision Letter 4]

22 Apr 2026

Spatial Disparities and Multilevel Determinants of Childhood Diarrhea in Mozambique: Evidence from the 2022–2023 Demographic and Health Survey (DHS)

PONE-D-25-40831R4

Dear Dr. Yewodiaw,

We’re pleased to inform you that your manuscript has been judged scientifically suitable for publication and will be formally accepted for publication once it meets all outstanding technical requirements.

Kind regards,

Khin Thet Wai, MBBS, MPH, MA

Academic Editor

PLOS One
---

## [Editor Report · Acceptance letter]

PONE-D-25-40831R4

PLOS One

Dear Dr. Yewodiaw,

I'm pleased to inform you that your manuscript has been deemed suitable for publication in PLOS One. Congratulations! Your manuscript is now being handed over to our production team.

Kind regards,

on behalf of

Dr. Khin Thet Wai

Academic Editor

PLOS One